# Diagnostic Approaches to Vascular Injury in Polytrauma—A Literature Review

**DOI:** 10.3390/diagnostics13061019

**Published:** 2023-03-07

**Authors:** Vuyolwethu C. Ntola, Timothy C. Hardcastle

**Affiliations:** 1Department of Surgical Sciences, Nelson R Mandela School of Clinical Medicine, University of KwaZulu-Natal, Durban 4058, South Africa; 2Trauma and Burns Service, Inkosi Albert Luthuli Central Hospital, Durban 4058, South Africa

**Keywords:** diagnostic approach, polytrauma, vascular injury, limb loss

## Abstract

Background: Polytrauma is understood as significant injuries, occurring at the same time, to two or more anatomical regions (the ISS regions) or organ systems, with at least one of the injuries considered as posing a threat to life. Trauma is the main cause of unexpected demise in individuals below the age of 44 years and represents a huge burden on society. Vascular injury is highly morbid; it can lead to rapid exsanguination and death, posing a threat to both life and the limb. Independent predictors of outcome include mechanism of injury, associated injuries, and time from injury to definitive care. The mechanisms of vascular injury in the setting of polytrauma are either blunt, penetrating or a combination of the two. Methods: Comprehensive literature review of current diagnostic approaches to traumatic vascular injury in the context of polytrauma. The factors influencing the diagnostic approach are highlighted. The focus is the epidemiology of vascular injury and diagnostic approaches to it in the context of polytrauma. Results: Traumatic vascular injuries are associated with limb loss or even death. They are characterised by multiple injuries, the dilemma of the diagnostic approach, timing of intervention and higher risk of limb loss or death. The systematic approach in terms of clinical diagnosis and imaging is crucial in order save life and preserve the limb. The various diagnostic tools to individualise the investigation are discussed. Conclusion: This paper highlights the significance of timely and appropriate use of diagnostic tools for traumatic vascular trauma to save life and to preserve the limb. The associated injury also plays a crucial role in deciding the imaging modalities. At times, more than one investigation may be required.

## 1. Introduction

In South Africa (SA), there are over 30,000 trauma-related deaths annually, which is equivalent to that of two-thirds of the whole trauma death rate for Europe. SA contributes to the increasing burden of road traffic collision (RTC)-related mortality [1]. Trauma is the leading cause of unexpected mortality in persons under the age of 44 years and represents a huge burden on society in terms of money spent for immediate and long-term care. Vascular injury is highly morbid, and under certain conditions it can lead to rapid exsanguination and death [2,3]. To avoid loss of life or limb loss, diagnostic investigations should be undertaken synchronously with the resuscitation process. The experience gained during periods of war has made limb salvage a rule rather than an exception. For a successful outcome, teamwork, communication among caregivers, sound knowledge of vascular anatomy and understanding of the appropriate sequence of diagnostic modalities are crucial [4]. Management of a polytrauma patient is very challenging, and the treating clinician is often faced with the dilemma of deciding which injury should be prioritised. Although rare, aortic/brain injury management can be challenging where there is a need to anticoagulate the patient but brain injury precludes the anticoagulation. Historically, these injuries have been associated with poor prognosis. However, technological advances such as endovascular aortic stenting, and the fact that anticoagulation is not absolutely contraindicated in traumatic brain injury, means many patients can survive with a good quality of life [5].

## 2. Clinical Diagnosis of Vascular Injury

The main clinical manifestations of vascular injury are haemorrhage (commonly from truncal vascular injuries) and ischemia (typical of peripheral arterial injuries) and, to a lesser extent, stroke after cervical vascular damage. Damage to major vessels will lead to haemorrhage and will clinically manifest as hypotension or shock. Although the clinical examination may identify vascular trauma, there is a diagnostic dilemma if the bleeding source cannot be identified. The signs of vascular injury in the limbs can be classified as either “hard” or “soft” signs. Hard signs include the following: bleeding from the artery, absence of pulse, pulsation or haematoma, bruit or thrill and ischemia-related signs that are increasing in size. Soft signs include the following: bleeding at the scene, reduced pulse, non-expanding hematoma, injury close to a blood vessel or bone fracture. In the presence of hard signs, intervention (surgical or endovascular) is generally recommended, whereas soft signs indicate the need for further investigation. One must be aware that negative clinical examination does not rule out vascular trauma [6,7]. Proper clinical assessment can yield up to 80% of patients with vascular injury [8].

## 3. Diagnostic Tools

### 3.1. Ankle Brachial Index (ABI)

The ankle brachial index is the ratio (ABI) of ankle blood pressure to that of the upper limb. It is a valuable and simple test. The patient is placed in a supine position to obtain reliable results. A doppler probe and a sphygmomanometer are required. The sphygmomanometer is inflated proximal to the artery in question. The inflation is continued until the pulse ceases, then the sphygmomanometer is gradually deflated until the pulse returns (as measured on a hand-held Doppler); this is to measure the arterial systolic blood pressure [4,9]. History and physical examination are the most crucial elements in trauma patients; however, as an adjunctive measure, non-invasive testing can be used to assist with clinical decision making [10]. The highest blood pressure is used (either left arm or right arm). The pressure of the affected lower limb is measured. The blood pressure of the posterior tibialis artery or dorsalis pedis of the affected limb is measured, and the highest value is used. When the ratio is <0.9, this may indicate a vascular trauma. Positive results need to be confirmed by an additional method. It can also be used to determine perfusion after vascular repair [4,9,10]. The ABI is an important component of vascular trauma evaluation of lower-extremity injuries [10,11].

### 3.2. Plain Chest Radiography

The accessibility of chest radiography makes it the most common initial imaging used for chest trauma evaluation [12]. According to the trauma resuscitation unit management guidelines, chest radiography remains one of the first-line imaging modalities of the thorax for severely injured patients to detect pneumothorax, haemothorax and mediastinal vascular injury [13]. There are several features of anterior posterior (AP) projection chest X-ray that may be suggestive of aortic injury. Widening of the mediastinum may be suggestive of aortic injury but it is not specific since it may indicate bleeding from any other thoracic vessel. The thoracic cavity to mediastinum ratio at the aortic arch of more than 0.25 or an absolute mediastinum width of 8 cm are markers of thoracic vascular injury. Loss of aortic knob contour is the most reliable sign to indicate blunt aortic injury. Mediastinum widening accompanied by tracheal deviation to the left may be indicative of an innominate artery injury [12,13,14,15] (See Figure 1). The AP projection may overestimate these features, and normal chest X-ray does not rule out traumatic vascular injury [14]. The mechanism of injury (blunt or penetrating) and haemodynamic status of the patient are important in determining the approach.

With penetrating injury, there is an added advantage in the sense that it provides the probable tract location, whereas in a blunt injury, a high index of suspicion is required. In penetrating injury, all zone 1 injuries should be included in the investigations of possible thoracic vascular injury [16].

### 3.3. Extended Focused Assessment with Sonar in Trauma (eFAST)

eFAST may be used to exclude haemothorax, pneumothorax, cardiac tamponade or abdominal injury. It is performed in the resuscitation bay [16]. This modality of ultrasound examines the abdomen and pericardium and can also image the anterior lungs bilaterally. This type of care test is well suited for assessment of a haemodynamically unstable trauma patient. The development of portable high-resolution ultrasound machines has enabled clinicians to assess the basic haemodynamic status of the trauma patients and thereby plan and prioritise interventions [16,17]. Many institutions have introduced the eFAST protocol in their trauma algorithm for primary survey. The eFAST exam has a sensitivity of 90–95% compared to 42–98% for traditional Focused Assessment with Sonography in Trauma (FAST), which excludes imaging the anterior lungs [18]. In experienced hands, eFAST provides a rapid and sensitive evaluation for haemothorax [19]. When used with limited cardiac sonogram, it can assess the nature and extent of associated injuries [16].

### 3.4. Duplex Sonography (DUS)

Duplex sonography (DUS) is widely used in the setting of trauma. Accessibility, low cost, easy mobility, non-invasiveness and lack of ionising radiation are its advantages. Luminal narrowing, hypoechoic intramural haematoma, dissected arterial wall, haemodynamic relevant stenosis, the “yinyang” sign (pseudoaneurysm) and occlusion are the features consistent with vascular injury. With the addition of colour-Doppler, the sensitivity, specificity and accuracy are up to 95%, 99% and 98%, respectively, in assessment of peripheral vascular injuries. It has the following limitations: missed injuries due to bony structures, open wounds, large haematomas, bulky dressings or splints, hindered access to the ultrasound probe and the fact that this test cannot be performed on patients with open wounds [20]. DUS is not reliable as a screening tool for blunt carotid vascular injury and cannot be used routinely for penetrating neck zone I and III injuries [7].

### 3.5. Echocardiography

Due to the good imaging quality of the pericardium and ventricles, transthoracic echocardiography (TTE) and transesophageal echocardiography (TEE) have a role in thoracic trauma. The dual capability of diagnostic information and haemodynamic monitoring is a key advantage of TEE; it assesses the hemodynamic status in relation to the cardiac filling and ventricular contractility, providing valuable information regarding the use of vasopressors. The following diagnostic information can be obtained: cardiac tamponade, segmental wall motion abnormalities, valvular pathology, as well as ascending and descending aorta injury [21]. Multiple TEE is found to be sensitive for detection of traumatic lesions involving the intimal and medial layers of the thoracic aorta. Helical CT and TEE have similar accuracy in the diagnosis of acute traumatic aortic injury. The use of these investigation modalities may be determined by availability and patient factors. In patients with suspected thoracic aortic injury with associated injuries such as rib fractures with flail chest, pulmonary contusion and extra thoracic trauma, a helical CT should be used as the screening imaging. In patients with an unremarkable mediastinum who sustained significant blunt chest trauma, a multiplane TEE should be used because of its greater sensitivity for the detection of superficial acute traumatic aortic injuries and blunt cardiac trauma. TEE may be considered for unstable patients with blunt chest injury for whom the transfer to the imaging suite may result in deterioration [22]. Haemorrhagic shock is common in polytrauma patients, and early recognition is fundamental to prevent mortality. TEE is valuable to identify the source of shock and also to provide information regarding the ventricular volume status [23].

### 3.6. Computed Tomography Angiography (CTA)

Newer multi-detector CT scanners have largely replaced digital subtraction angiogram, which had been previously regarded as the gold-standard imaging modality for vascular injury. CTA has a high sensitivity and specificity, with negative predictive value of 97–100% for diagnosis of neck vascular injuries. The CTA has the added advantage of being non-invasive and allows for assessment of other related injuries in a polytrauma patient, although it is less accurate for aero-digestive injury. The simultaneous cervical CTA with whole body CT scan is possible; however, there have been reports of missed blunt carotid vascular injuries. This imaging modality allows for grading of injury and appropriate planning for intervention. In penetrating injury, the CTA has the ability of delineating the trajectory of the missile in relation to the vital organs. Penetrating neck injuries that breach the neck fascia should be investigated via CTA regardless of the anatomical zone [4,9].

For trauma patients, the primary imaging technique is the CT scan. The fact that it provides a superior spatial resolution, allowing quick acquisition and rapid reconstruction times which are crucial in providing definitive care in the “Golden Hour”, makes it the imaging modality of choice. Consequently, it has become the primary imaging technique [10,20].

CTA has favourable features for diagnosis of aortic injuries. It has a sensitivity of 86–100% and a specificity of 40–100% and, most importantly, a negative predictive value of 99.3–100%. Among the commonly injured vessels are those of the extremities, and this pattern is observed in both the civilian and the military settings. Clinical examination is key for early detection of vascular injury. In stable patients with non-conclusive clinical findings, CTA has become the chief imaging modality for screening and diagnosis and for potentially planning intervention [24].

In patients with suspected abdominal vascular trauma who do require immediate exploratory laparotomy, most algorithms suggest CT as a standard investigation. It is used for both penetrating and blunt abdominal trauma, provided the patient is stable enough to reach the scanner. It has a sensitivity of 95–100% and a specificity of 87–100% for evaluation of femoropopliteal and mid–proximal upper limb vascular injury. Foreign body artefacts rarely interfere with the interpretation of the imaging [4].

Since it is relatively available, less invasive and reduces unnecessary exploration, CTA is becoming the diagnostic tool of choice during initial evaluation of stable patients with vascular injury [25]. CT scanning has resulted in an increased detection of occult abdominal vascular injuries in the setting of blunt abdominal trauma, especially those involving the renal arteries where thrombosis and intimal flaps are the most common lesions.

CTA was reputed to be associated with contrast-induced nephropathy and increasing use of imaging and interventions that require intravascular contrast media administration in Sub-Saharan Africa, which may have resulted in an increased number of patients developing contrast-induced nephropathy [3,26]. Due to variable definition and lack of control group, the incidence is unclear. It is reported as the third leading cause of iatrogenic renal insufficiency. Hypotension, resuscitation, severe tissue injury and direct kidney injury are the factors that increase the risk of contrast-induced nephropathy. These factors are observed in polytrauma patients.

Since the definition of contrast-induced nephropathy requires the exclusion of the other factors as the cause of renal insufficiency, it makes it difficult to diagnose contrast-induced nephropathy in trauma patients. There is, however, growing evidence that administration of intravascular contrast media has very little consequence on renal function, and literature is conflicting for trauma patients. It has been found that patients who received additional contrast within 72 h of their initial dose did not have increased odds of developing acute kidney injury. This is an important finding as the imaging studies and angiography far outweigh the risk of repeat contrast exposure within polytrauma patients [27]. Several recent large studies conclusively show no contrast nephropathy risk in most emergency imaging studies. The consensus is that there is no real risk of contrast-induced nephropathy [28,29].

### 3.7. Magnetic Resonance Imaging/Angiography (MRA)

MRA has been used in certain patients with blunt carotid vascular injury; however, the results were variable. It has several major limitations that rule out this imaging modality as a desirable imaging tool in the emergency setting. Some of the limitations are as follows: requirement for MR-compatible equipment (especially for ventilated, sedated polytrauma patients), unavailability out of normal working hours, long image processing times and the potential contra-indication in patients with metallic implants or foreign bodies. For patients who require simultaneous assessment of the brain and spinal injury, it could be of great benefit. It is of great benefit for early recognition of early cerebral ischaemia, which is very important in blunt cerebrovascular injury. Biffl et al. reported compared MRA with catheter angiography and reported a sensitivity of 75% and a specificity of 67%. Corresponding positive predictive and negative predictive values were 43% and 89%, respectively. Miller et al. revealed even lower figures for 21 patients. Pseudoaneurysms are known to be missed due to turbulent flow. The role of MRI/MRA thus remains supportive rather than conclusive in the context of the polytrauma patient [30,31].

### 3.8. Digital Subtraction Angiography (DSA)

DSA is a fluoroscopic technique usually used for precisely outlining the blood vessels by subtracting radiopaque structures such as bone. It enables simultaneous diagnosis and intervention, saving time [32]. It is an invasive procedure with the following complications: puncture site bleeding and haematoma, unintended vessel puncture leading to arteriovenous fistula and complications related to embolism. It requires mobilisation of theatre personnel, and it is performed in specialised centres. Although this imaging has been regarded as the gold-standard imaging modality for vascular trauma, in the modern practice, its role has been restricted to equivocal cases after CTA [33]. DSA has a high sensitivity and is preferred for diagnosis of penetrating cerebrovascular injuries (PCVI) since it has been demonstrated that CTA has lower sensitivity, specificity and positive and negative predictive values for the diagnosis of PCVI cerebrovascular injuries [34] (see Figure 2). This imaging modality is also performed as part of endovascular intervention or in patients with foreign objects interfering with CTA interpretation. Timing is important, and the balance between diagnosis and life-over-limb must be considered when compared to open surgery.

### 3.9. Intra-Vascular Ultrasound (IVUS)

IVUS, first developed in the 1960s, provides real-time 360° vessel images using a miniature ultrasound probe. The limitation of CTA is high sensitivity but lower specificity. Patients with unclear CTA results often require additional imaging, including aortography, IVUS or transesophageal echocardiography. IVUS has an outstanding accuracy for diagnosis of traumatic aortic injury. It significantly impacts on the endograft sizing for endovascular management of traumatic aortic injury and reduces device-related complications. IVUS has the following advantages: it does not use contrast, there is no associated radiation exposure, and it uses the same access site as the angiography. IVUS has the following disadvantages: it is not available in most centres in developing countries, it incurs additional costs, it has a larger sheath, and it requires longer operating room times [35].

## 4. Approach to and Use of on-Table Imaging

On-table imaging offers an added advantage because it allows patients to receive diagnostic and therapeutic procedures in the same setting with no delays, and these procedures are crucial in polytrauma patients, especially those who are rapidly transferred to the operating suite due to instability and lack of opportunity to obtain further imaging. The multifunctional image-guided therapeutic suites (MIGTS) combine standard high-quality imaging with an operating room. This “hybrid” approach circumvents the risk and delay from additional transport within the hospital and allows swift switches between diagnostic and sterile surgical procedures [36].

The role of interventional radiology is crucial in polytrauma patient management. For full application of on-table imaging services, the following are required: excellent digital subtraction angiographic apparatus, ideally with digital road mapping and/or fade-fluoroscopic capabilities; well-trained staff and apparatus needed for monitoring of very ill patients; and ability to read these resources immediately [37].

In less well-equipped facilities or in a dire emergency, a simple C-arm or plain-film with hand-injected contrast can assist in vascular assessment of a patient who was too unstable to undertake more complex imaging, and it can also assist in visualising a successful on-table repair. While used less often today, this remains an emergency option in the polytrauma patient with a potentially threatened limb [38].

## 5. Differences in Approach to Blunt vs. Penetrating Injuries

There remain important differences to be highlighted between blunt and penetrating injuries when it comes to the diagnostic approach.

### 5.1. Neck Trauma and Chest Trauma

The modern paradigm of imaging penetrating neck trauma favours MDCTA if there are no hard signs warranting immediate exploration. If the MDCTA is inconclusive or in cases of high clinical suspicion, further evaluation with fluoroscopic oesophagography may be required. In blunt neck trauma, patients should be screened with MDCTA to rule out cerebrovascular injuries, which may provide an opportunity to rule out the very rare occurrence of pharyngoesophageal perforation [24,38]. See Figure 3 for a useful flow diagram of the algorithm for investigation.

For either blunt or penetrating chest trauma, the following imaging modalities can be used: Chest X-ray, eFAST, Echocardiography and CT chest, which are all important depending on the availability. See Figure 4 for the chest trauma approach.

### 5.2. Abdominal Trauma

For blunt abdominal trauma, in the absence of a need for emergency laparotomy, imaging is used to determine if emergency laparotomy is required. Abdominal X-ray has a very limited role in the evaluation of blunt abdominal trauma [39]. eFAST should be used instead as it can be performed quickly and provides immediate results; however, it requires more than 250 mL of fluid to be collected in Morrison’s pouch to yield positive results, and it does not identify the injured anatomical structure while remaining operator-dependent. Due to the advent of eFAST, diagnostic peritoneal lavage is rarely performed in modern practice. For a stable patient, abdomen and pelvis MDCTA is performed. It localises specific anatomic areas that are injured, and it allows grading of the injuries which guide in the planning of intervention. However, it requires the patient to leave the emergency department, and there a risk of contrast-induced nephropathy and radiation exposure [40].

For penetrating injury, two key variables should be established: whether the wound penetrates the peritoneum and whether there is intra-peritoneal injury. The presence of free air under the diaphragm seen on a chest X-ray may confirm peritoneal penetration. Abdomen ultrasound may confirm the presence of free fluid in the abdomen or evidence of violation of the fascia; however, it is not sensitive. Abdomen CT with IV contrast is the optimal method for determining both the intra-peritoneal penetration and intra-peritoneal injury where emergency laparotomy is not required [41].

In abdominal trauma, there is currently limited available reliable scientific proof of the role for diagnostic laparoscopy as a standard diagnostic procedure. In the literature, there is conflicting information regarding the application of this procedure. The European Association for Endoscopic Surgery accepts that laparoscopy may be taken into consideration in diagnostics of patients after abdominal trauma; however, the 2010 publication by the Journal of Trauma points out that this may be considered only in post-trauma diagnostics of intra-peritoneal penetration and suspicion of diaphragm injuries [42]. Koto et al. established the diagnostic laparotomy feasibility and safety in trauma. For identification of intra-abdominal injuries, it is precise. To reduce the rate of missing injuries, the following are required: preoperative imaging, compliance to the pre-defined steps of procedure and excellent systematic laparoscopic assessment [43].

#### Extremity Trauma

ABI is a useful tool for the initial evaluation of suspected vascular trauma of the lower extremity. Preexisting peripheral vascular disease may render it less reliable; thus, positive results must always be confirmed with additional imaging [6]. The presence of hard signs in a patient with penetrating injury carries a recommendation for immediate intervention. There are some exceptions where the patient may require imaging to localise the injury, such as multiple bone fractures and gunshot wounds with the trajectory of the bullet along the long axis of the limb [44]. Duplex ultrasound is useful for evaluation of extremity vascular injury; however, it requires experienced staff to operate it, while patients with larger wounds or with injuries in multiple areas may not be good candidates. Duplex Doppler ultrasound plays a major role in the diagnosis of both trauma and blunt extremity vascular injury, and it has a sensitivity of more than 94%. Many trauma centres have included the ultrasound and color flow duplex points of care as tools for investigation of polytrauma patients; however, high BMI, haematoma, subcutaneous emphysema and large open wounds must be kept in mind [44,45]. The development of MDCT has led to CTA being the most performed imaging for evaluation of suspected extremity vascular injury. Its performance is compared to conventional angiography. Rapid image acquisition, 3D reconstruction and being able to assess the adjacent structures are the added advantages [46]. Conventional angiography has largely been replaced by the other imaging modalities; however, it is still regarded as the gold standard. The advantage of this imaging is that an intervention can be performed on the same setting; however, some authors believe that it can lead to an increase in the ischemic time. This imaging should be only considered if the other imaging modalities are inconclusive in patients for whom there is a high index of suspicion of vascular injury. Complications are related to the puncture sites, such as pseudoaneurysm, haematoma and fistula; or to the diagnostic procedure itself, such as clot embolism leading to acute limb ischemia [47]. Whenever one approaches these patients, a highly individualised approach should be practiced.

## 6. Haemodynamically Stable Patient vs. Untable Patient

The approach to the two categories of patients (haemodynamically stable and unstable patients) is different. In the case of a stable patient, CTA or other noninvasive investigation such as doppler ultrasound, depending on the area of concern and access to available resources, are the imaging modalities of choice. These patients are stable and there is time for further investigation [24]. For unstable patients, after immediate emergency room bleeding control, the best option will be immediate exploration, without the need for imaging. The other category of patient that requires immediate exploration without the need for imaging are patients with peripheral vascular injury and signs of ischemia [47]. For unstable patients, the use of on-table angiography followed by immediate endovascular intervention is also a recognised approach, provided that it will not introduce delays to either haemorrhage control or repair.

## 7. Conclusions

This paper highlights the significance of the timely and appropriate use of diagnostic tools for traumatic vascular trauma to save life and to preserve the limb. The following factors determine the imaging modality to be used: patient factors (haemodynamic stability, injured anatomical area and associated injuries), cost and availability. At times, more than one investigation may be required. It has also been noted that advances in imaging have a direct impact upon patient management.

A very clear example is the case of a mandatory neck exploration following penetrating neck injury progressing over time to a selective non-operative approach. Although the DSA is a gold standard, its use has now been restricted only to patients with equivocal CTA results. On the other hand, the introduction of endovascular interventions has also resulted in increased utilisation of therapeutic DSA.

It remains a clinician’s responsibility to carefully determine the optimal diagnostic test for a patient with a suspected vascular injury, and that requires in-depth knowledge of all the available diagnostic images, along with multi-disciplinary team-based interaction to determine safe timing and balance between testing and intervention for vascular and other injuries in the context of polytrauma.

## Figures and Tables

**Figure 1 diagnostics-13-01019-f001:**
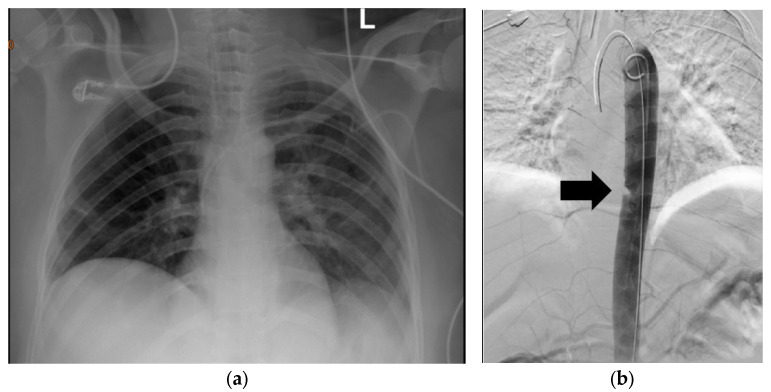
(**a**) Plain chest X-ray of a 40-year-old who was involved in a motor vehicle collision. The patient had fractures in ribs 4–8. (**b**) DSA demonstrating descending aortic injury. The arrow indicates a filling defect.

**Figure 2 diagnostics-13-01019-f002:**
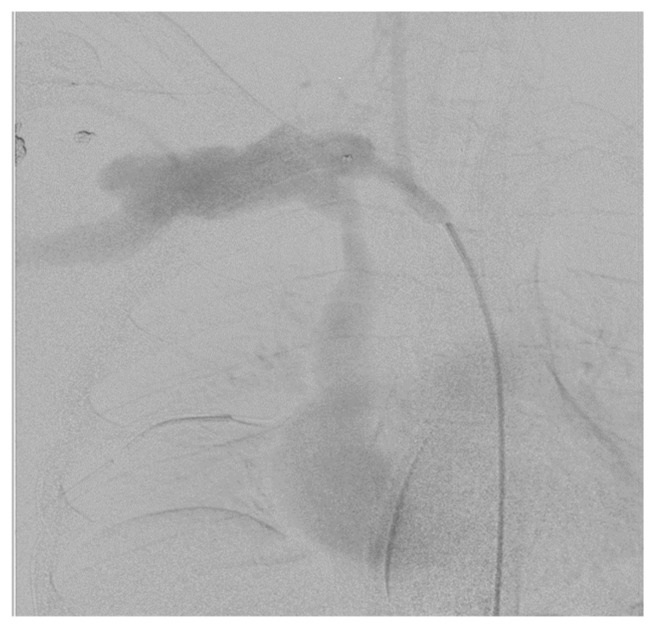
DSA of a 24-year-old male who sustained a zone I gunshot wound on the right side. DSA showed injury of the distal part of the right subclavian artery, and the patient was subsequently managed via open repair using a prosthetic graft.

**Figure 3 diagnostics-13-01019-f003:**
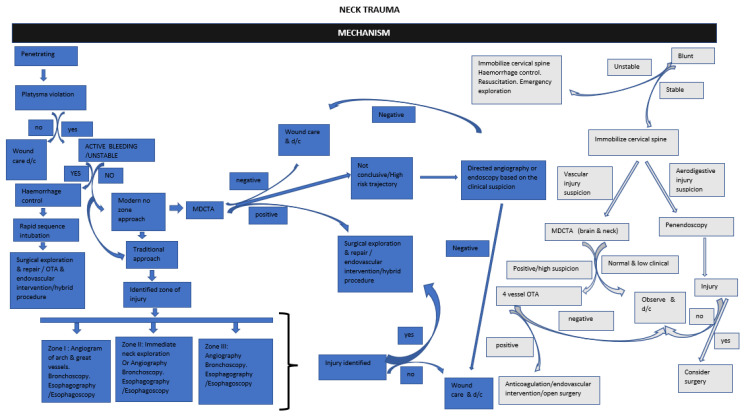
Suggested algorithm for approach to neck trauma. d/c−discharge; MDCTA−Multidetector Computed Tomographic Angiography.

**Figure 4 diagnostics-13-01019-f004:**
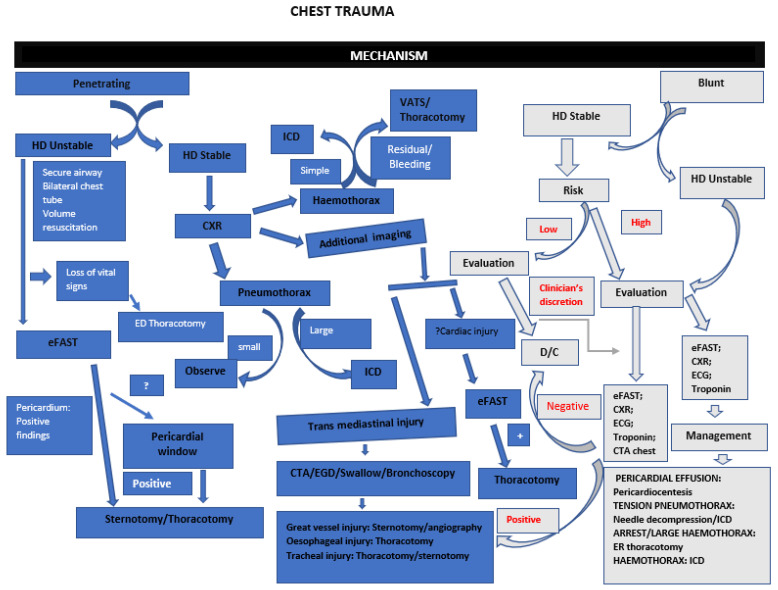
Suggested algorithm for approach to chest trauma. HD—Haemodynamically; ED—Emergency department; eFAST—Extended focused assessment with sonography for trauma; “?”—Not conclusive; CXR—Chest X-ray; ICD—Intercostal chest drain; VATS −Video-assisted thoracoscopy; CTA—Computed tomography angiogram; EGD—Esophagogastroduodenoscopy.

## Data Availability

The study did not report on any new data.

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
