# Peer review of "Diagnostic Approaches to Vascular Injury in Polytrauma—A Literature Review"

_diagnostics, 2023, doi:10.3390/diagnostics13061019_

Round 1
Reviewer 1 Report
The present work made a comprehensive review on the diagnosis of all vascular problems that may arise in polytraumatized patients. To do so, the authors investigated all forms of vascular diagnosis and health with multiple current literature sources. As a result, they observed that multiple tools exist to measure vascular damage.
The study has several strengths, which are the up-to-date literature citations; the description of the different tools and their use depending on the situation.
The study is original, well conducted and well presented. The results are solid and carry a relevant message for correct diagnosis.
I have no specific requests for the authors.
Author Response
Reviewer 1: Thank you for your feedback. The was no added suggestion
Reviewer 2:
- There is a lack of reference to peripheral vascular injuries in the lower and upper limbs. Even the CTA paragraph does not mention the importance of diagnosing these injuries.
We added the section addressing the approach to extremity vascular injury (lower & upper limb)
- What is the reason that Figure 3 was attached? Why attach an algorithm detailing the approach to diagnosing neck injuries? What about chest injuries? Abdominal injuries? The chapter on the comparison between blunt and penetrating injuries requires restructuring and perhaps sub-chaptering each body area with the challenges and tips relevant to the diagnostic tools.
We have included a figure detailing the approach to chest trauma. We did not attach a flow-chart figure on extremity and abdominal trauma as it is extensively explained in the respective text section and we are limited by the amount of figures allowed by the journal.
- I recommend adding a chapter that discusses the choice of diagnostic tool based on whether the casualty is hemodynamically stable or unstable, similar to the chapter on penetrating injury versus blunt injury.
A section on haemodynamically unstable or stable patients was added, with appropriate discussion and additional references
Reviewer 2 Report
Thank you for the opportunity to read and comment on your review. The topic "Diagnostic approaches to Vascular injury" is a significant challenge in treating trauma casualties and saving their lives. The right choice of diagnostic means at the right time is the key to success in treating vascular injuries. The work summarizes well the current information in the medical literature. Several recommendations and minor comments:
1. There is a lack of reference to peripheral vascular injuries in the lower and upper limbs. Even the CTA paragraph does not mention the importance of diagnosing these injuries.
2. What is the reason that Figure 3 was attached? Why attach an algorithm detailing the approach to diagnosing neck injuries? What about chest injuries? Abdominal injuries? The chapter on the comparison between blunt and penetrating injuries requires restructuring and perhaps sub-chaptering each body area with the challenges and tips relevant to the diagnostic tools.
I recommend adding a chapter that discusses the choice of diagnostic tool based on whether the casualty is hemodynamically stable or unstable, similar to the chapter on penetrating injury versus blunt injury.
Author Response
Reviewer 2:
- There is a lack of reference to peripheral vascular injuries in the lower and upper limbs. Even the CTA paragraph does not mention the importance of diagnosing these injuries.
We added the section addressing the approach to extremity vascular injury (lower & upper limb)
- What is the reason that Figure 3 was attached? Why attach an algorithm detailing the approach to diagnosing neck injuries? What about chest injuries? Abdominal injuries? The chapter on the comparison between blunt and penetrating injuries requires restructuring and perhaps sub-chaptering each body area with the challenges and tips relevant to the diagnostic tools.
We have included a figure detailing the approach to chest trauma. We did not attach a flow-chart figure on extremity and abdominal trauma as it is extensively explained in the respective text section, also to note we are restricted by the journal rules for the number of images/figures
- I recommend adding a chapter that discusses the choice of diagnostic tool based on whether the casualty is hemodynamically stable or unstable, similar to the chapter on penetrating injury versus blunt injury.
A section on haemodynamically unstable or stable patients was added, with appropriate discussion